# Engineering Peptide Inhibitors of the HFE–Transferrin Receptor 1 Complex

**DOI:** 10.3390/molecules27196581

**Published:** 2022-10-04

**Authors:** Daniela Goncalves Monteiro, Gautam Rishi, Declan M. Gorman, Guillaume Burnet, Randy Aliyanto, K. Johan Rosengren, David M. Frazer, V. Nathan Subramaniam, Richard J. Clark

**Affiliations:** 1School of Biomedical Sciences, The University of Queensland, Brisbane, QLD 4072, Australia; 2Centre for Genomics and Personalised Health, School of Biomedical Sciences, Queensland University of Technology (QUT), Brisbane, QLD 4059, Australia; 3The QIMR Berghofer Medical Research Institute, 300 Herston Rd, Brisbane, QLD 4006, Australia

**Keywords:** hepcidin, stapling, HFE, TFR1

## Abstract

The protein HFE (homeostatic iron regulator) is a key regulator of iron metabolism, and mutations in HFE underlie the most frequent form of hereditary haemochromatosis (HH-type I). Studies have shown that HFE interacts with transferrin receptor 1 (TFR1), a homodimeric type II transmembrane glycoprotein that is responsible for the cellular uptake of iron via iron-loaded transferrin (holo-transferrin) binding. It has been hypothesised that the HFE/TFR1 interaction serves as a sensor to the level of iron-loaded transferrin in circulation by means of a competition mechanism between HFE and iron-loaded transferrin association with TFR1. To investigate this, a series of peptides based on the helical binding interface between HFE and TFR1 were generated and shown to significantly interfere with the HFE/TFR1 interaction in an in vitro proximity ligation assay. The helical conformation of one of these peptides, corresponding to the α1 and α2 helices of HFE, was stabilised by the introduction of sidechain lactam “staples”, but this did not result in an increase in the ability of the peptide to disrupt the HFE/TFR1 interaction. These peptides inhibitors of the protein–protein interaction between HFE and TFR1 are potentially useful tools for the analysis of the functional role of HFE in the regulation of hepcidin expression.

## 1. Introduction

Hepcidin is a peptide composed of 25 amino acid residues inter-linked by four disulfide bonds [1]. This hepatocyte-secreted hormone is the central regulator of systemic iron metabolism as it negatively controls circulating iron levels by inhibiting iron absorption from the diet, and iron mobilisation from macrophagic and hepatic stores [2]. Hepcidin functions by binding to the sole known mammalian iron exporter, ferroportin, thus inducing its internalisation and subsequent lysosomal degradation, ultimately inhibiting iron efflux into the bloodstream [3]. The hepcidin–ferroportin axis is central to systemic iron homeostasis and a dysregulation of hepcidin results in two phenotypically contrasting groups of clinical conditions: one in which increased hepcidin levels cause or contribute to iron-restrictive anaemias, and the other characterised by hepcidin deficiency, which results in iron overload, as observed in nearly all forms of hereditary haemochromatosis (HH) and iron-loading anaemias such as β-thalassaemia and congenital dyserythropoietic anaemias [4].

Although mutations in the hereditary haemochromatosis protein HFE (homeostatic iron regulator) underlie the most frequent form of HH (HH-type I), the molecular function of this protein is still largely unclear [5]. HFE is a type I transmembrane glycoprotein homologous to class I major histocompatibility complex (MHC) molecules and associates with β2-microglobulin (β2m) [6]. A series of structural and mutagenesis studies have shown that HFE interacts with transferrin receptor 1 (TFR1), a homodimeric type II transmembrane glycoprotein that is responsible for the cellular uptake of iron via iron-loaded transferrin binding (Figure 1) [6,7]. These studies indicate that the binding interface between HFE and TFR1 involves the HFE helical domains α1 and α2 and the helical region of TFR1, which has also been implicated in transferrin binding (Figure 1) [6,8]. Furthermore, the complex appears to have two-fold symmetry with 2:2 stoichiometry.

Mutant mouse strains with constitutive binding of HFE to TFR1 develop iron overload, whereas mice carrying a mutation that inhibits HFE/TFR1 complex formation develop iron deficiency as a result of inappropriately high hepcidin expression [9]. This suggests that HFE is involved in a sensory system for circulating iron levels, which may rely on a competitive mechanism between HFE and holo-transferrin binding to TFR1. It has been speculated that at high concentrations of holo-transferrin, HFE displacement from TFR1 binding leads to the formation of a complex comprising HFE, transferrin receptor 2 (TFR2), and maybe hemojuvelin (HJV), which functions to potentiate the main regulatory pathway for hepcidin expression in hepatocytes involving the bone morphogenetic protein (BMP) receptor [10,11]. However, this potential regulatory mechanism is still under investigation as several studies have failed to observe an interaction between HFE and TFR2 [12,13,14]. HFE has been found to interact with Alk3, a type I serine threonine kinase receptor subunit of BMP complexes and to inhibit its ubiquitination and proteasomal degradation, resulting in an increase in Alk3 protein expression and accumulation at the cell surface [15]. This stabilising function is thought to underlie the potentiating effect of HFE on BMP signalling and SMAD1/5/8 phosphorylation, and concomitant increase in hepcidin expression.

According to a statistical analysis of data from the Protein Data Bank (PDB), approximately 62% of the protein–protein interactions appear to involve α-helical interfaces, often ranging from 1 to 4 helical turns, which corresponds to around 4–15 amino acid residues [16]. However, under physiological conditions and in the unbound state, short helical fragment peptides do not tend to form thermodynamically stable α-helical structures, as water molecules compete with the polar backbone amide groups for hydrogen bonding, thus hindering the backbone hydrogen bonds required for α-helicity [17]. Therefore, several methods have been utilised to stabilise synthetic peptides in α-helical arrangements, including the incorporation of salt bridges, chelating metal ions, or covalent linkers that cyclise peptide segments [18].

Here, we show that peptides based on the interacting sequence of either HFE or TFR1 can successfully disrupt complex formation between HFE and TFR1 in vitro. These would be of use not only to investigate HFE-mediated signalling but also as potential lead molecules for developing therapeutics for iron loading disorders. Based on this finding, second-generation analogues of one of the most active peptides of the series, HFE α1α2, were developed, by incorporating α-helix stabilising lactam linkages with the goal of enhancing efficacy. These second-generation peptides showed increased helical propensity, but this did not translate to improved activity relative to the unstapled peptide.

## 2. Results and Discussion

### 2.1. Rational Design of Peptide Inhibitors of the HFE/TFR1 Complex

Protein–protein interactions are commonly mediated by α-helical segments that offer a plethora of highly valuable therapeutic targets. Peptides, in comparison to small molecule therapeutics, offer a level of potency and selectivity for their drug targets that successfully enable the disruption of more extensive interactions. In order to propel the investigation of the molecular role of HFE and of its complex formation with TFR1, we designed a library of peptides to target this protein–protein interaction. This library is composed of two families of peptides, each based on the interacting surface of HFE and TFR1, which primarily involve the helical domains α1 and α2 of HFE and two helices within the helical region of TFR1 (Figure 1).

In the crystal structure of the HFE/TFR1 complex, HFE α1 helix appears to interact with TFR1 helix 3 in an antiparallel fashion over nearly the entire lengths of both helices, corresponding to approximately five helical turns. TFR1 helix 1 is slightly shorter than the other TFR1 helices in the bundle and is thought to interact with the C-terminal region of the HFE α1 helix in a parallel arrangement. Despite the smaller interacting surface, this helical pair forms a hydrophobic core consisting of TFR1 Leu^619^ and Val^622^, and HFE Val^78^ and Trp^81^, as determined by site-directed mutagenesis studies and clinical HFE mutations associated with HH (Ile^83^Thr and Gly^71^Arg) [6,19]. In addition, residues in the α2 helix of HFE also interact with helix 3 of TFR1 helical domain, and Arg^629^ in helix 2 of TFR1 has a number of polar interactions with residues in the α1 and α2 helical domains of HFE [6]. Therefore, we designed a series of 10–30 residue peptides based on HFE α1 and α2 helical segments and helices 1 and 3 of TFR1 that incorporated the key amino acid residues involved in complex formation (Table 1). In addition, we altered some residues in the sequences to be consistent with the mouse homologues of TFR1 and HFE as our assay was to be performed on mouse liver cells. Negative controls were also designed using segment regions from outside of the proposed interacting surface or by scrambling the sequence of interest (Table 1).

### 2.2. Structural Features of HFE- and TFR1-like Peptides

Due to the helical nature of the protein regions used for the rational design of peptides targeting the interaction between HFE and TFR1, we predicted that the peptide fragments could adopt some level of helical secondary structure. To investigate this, we used ^1^H NMR assignment and random coil values to calculate the Hα secondary chemical shifts for each residue [20]. These shifts were then plotted together to provide an overview of the secondary structure of each peptide in aqueous media (Figure 2). Hα secondary chemical shift values closely around 0 (±0.1) commonly correspond to randomly structured regions, with stretches of positive and negative values suggesting β-sheet and α-helical arrangements, respectively [21].

Shorter peptides tend to adopt less well-defined structures than longer peptides and proteins, as folding into compact arrangements involves extensive stabilising intramolecular interactions. The two shorter peptides in the TFR1 series (h1 and h3) appear to be relatively unstructured, with Hα secondary chemical shifts being close to zero (Figure 2A). However, the longer TFR1 h1h3 peptide appears to adopt some helical character based on a series of negative secondary shift values for residues Leu^10/618^ to Leu^17/645^. This section includes a truncated sequence corresponding to h2 of TFR1 (Phe^12/631^ to Thr^14/633^), which could also contribute to its slightly helical nature. In addition, a series of HN_i_-HN_i + 1_ NOE crosspeaks in this region further supports the presence of helical structure. For the HFE series, the peptides have a stronger tendency for α-helical character, as shown by the negatively skewed pattern of Hα secondary chemical shifts and HN_i_-HN_i + 1_ NOE crosspeaks (Figure 2B). This is particularly noticeable for HFE α1 between Gly^9/97^ and Gly^23/111^ and less so for HFE α2 between Asp^3/182^ and His^5/184^. Interestingly, for HFE α1α2, which corresponds to a peptide containing helical segments from both α1 and α2 helical domains of HFE joined by a couple of glycine residues, the Hα secondary chemical shift analysis shows two regions of negative values that align with each of the α1 and α2 helical regions (Figure 3) although for these shifts are not as pronounced in the α1 region as in the HFE α1 peptide; this is presumably because HFE α1 is N-terminally extended when compared to HFE α1α2.

Overall, Hα secondary chemical shifts from ^1^H NMR assignment show that both HFE- and TFR1-like peptides have a tendency towards helical structure. This is an encouraging starting point as the binding between HFE and TFR1 is mediated by helical segments: α1 and α2 helices of HFE; h1, h3, and to a smaller extent h2 helices of TFR1. As a result, both sets of peptides could be expected to readily adopt favourable binding conformations and successfully simulate partial physiological interfaces for native HFE and TFR1.

### 2.3. HFE- and TFR1-like Peptides Inhibit HFE/TFR1 Complex Formation in an In Situ Proximity Ligation Assay

Both HFE- and TFR1-derived series of peptides were tested in vitro for their capacity to disrupt the interaction between HFE and TFR1. For this, a proximity ligation assay, Duolink^®^, was used. This has been shown to enable the detection, visualisation, and quantification of weak or transient protein–protein interactions in their native state and without having to overexpress the protein targets. The assay has previously been established for the detection of complex formation between HFE and TFR1 in mouse liver cells (Hepa 1–6), which endogenously express TFR1 [14].

Contrary to control peptides, all peptides targeting the binding interface significantly reduced the number of molecular interactions between HFE and TFR1 relative to untreated cells (Figure 3), as observed by the reduction in the number of bright red spots corresponding to HFE/TFR1 protein–protein interactions. The most active peptides were found to be TFR1 h1, h3, TFR1 h1h3, and HFE α1α2 (Figure 3), which significantly hinder complex formation relative to both untreated cells and cells treated with negative control peptides (HFE and TFR1 control peptides). The specificity of the HFE α1α2 peptide was also confirmed by testing an additional negative control of identical composition, but with a randomly scrambled sequence (HFE α1α2 scrambled). Interestingly, there was little difference in activity between the shorter peptides and the longer combined sequences for either the TFR1 or HFE series (Figure 3). TFR1 h3 appeared to be the most effective at inhibiting the HFE/TFR1 interaction. Based on the x-ray crystal structure this helical region appears to interact with a groove on the HFE interface, which is common binding mode for other α-helix mimics that inhibit protein–protein interactions [23].

### 2.4. Optimising HFE α1α2 by Helical Stapling

It is not unusual for helical motifs to lose their helicity when detached from their original full-length protein, and this is expected to negatively affect their binding affinity and, thus, their bioactivity. As a result, several methods for helical stabilisation have been developed, including producing a “stapled” peptide where proximate amino acid side chains on one side of a helix are crosslinked to stabilise the helical structure. A number of chemical staples have been explored including disulfide, thioester, diester, triazole, pyrazole, lactam, and hydrocarbon linkages, connecting residues i to i + 3, i to i + 4, or i to i + 7. Recently, lactam linkers established between Asp^i^ and Lys^i + 4^ have been found to confer the greatest α-helicity in water, in comparison to hydrocarbon, triazole, and thio- and dithioether alternatives [17].

With the aim of stabilising the helical propensity of HFE α1α2, to potentially improve its potency, we incorporated a lactam staple on the helical surface of the peptide opposite to the face that is expected to contact TFR1. We chose HFE α1α2 as, based on the NMR data (Figure 4), we believed that a stapling approach would have the biggest impact on the helicity of this peptide, given that HFE α1 showed significant helical character. For this, we generated helical projections using NetWheels [24] to map the key interacting amino acid residues included in the peptide motif and determined whether we could use any Lys or Asp residues from the parent sequence to form the staple. This was true for Asp^6/105^ and Lys^21/185^, which gave rise to two rationally designed single stapled derivatives: HFE α1α2-s1, where Trp^17/181^ was mutated to Asp and was lactam linked to Lys^21^, and HFE α1α2-s2, where Ile^10/109^ was mutated to Lys and linked to Asp^6^ (Figure 4 and Table 1).

In short, each portion of the linker was incorporated as Fmoc-Lys(Mtt)-OH or Fmoc-Asp(OPip)-OH for Lys and Asp residues, respectively, and following the incorporation of Fmoc-Asp(OPip)-OH at position 17 for HFE α1α2-s1 and at position 6 for HFE α1α2-s2, selective removal of the side-chain protecting groups and subsequent on-resin side-chain cyclisation was used to introduce the lactam staples between the respective sidechains. This process was repeated for the double stapled peptide (HFE α1α2-s1/2). The remaining amino acid residues in the sequence were then coupled, and the peptides were cleaved and purified as previously described.

### 2.5. Structural Analysis of the Stapled HFE α1α2 Analogues

Hα secondary chemical shifts were calculated for the stapled variants from the ^1^H chemical shift data for each peptide and were then compared to those of the parent peptide (Figure 4B). As previously remarked, stretches of largely negative (<−0.1 ppm) Hα secondary chemical shifts are often indicative of α-helical regions [21]. For both HFE α1α2-s1 and HFE α1α2-s2, the amino acid residues around the linkage experience significant deviations towards more negative values relative to the parent peptide (see Asp^18/182^ for HFE α1α2-s1 and Phe^7/106^ for HFE α1α2-s2, Figure 4B). Nevertheless, away from the stapled region, the Hα secondary chemical shifts for both derivatives are identical to those of HFE α1α2, suggesting that stapling only impacts structural features at a local level.

### 2.6. HFE α1α2-Single Stapled Analogues Are of Similar Potency as HFE α1α2 at Disrupting HFE/TFR1 Complex Formation in an In Situ Proximity Ligation Assay

The capacity of the HFE α1α2 stapled analogues to disrupt complex formation between HFE and TFR1 was subsequently investigated using the same in vitro proximity ligation method previously described. Interestingly, both single-stapled derivatives appear to be just as potent as the parent peptide, resulting in a similar number of HFE/TFR1 interactions detected as bright red spots (Figure 5). Quantification of this effect was again carried out using the “Find Maxima” function of FIJI across multiple images. The double stapled analogue (HFE α1α2-s1/2) also prevents HFE/TFR1 interactions but is not significantly different to either the single-stapled or unstapled HFE α1α2, indicating that the addition of further structural rigidity does not infer improved inhibitory activity (Figure 5).

## 3. Conclusions

In summary, we have successfully shown that the HFE/TFR1 interaction can be targeted by rationally designed peptides from the binding interface of either HFE or TFR1. Further studies in other assay systems are required to validate our findings, but it is hoped that these peptides will be useful tools to help uncover the biological effects of modulating the binding interaction between HFE and TFR1. It should be noted that the binding interface between HFE and TFR1, from which we have designed these peptides, appears to overlap with that of transferrin and TFR1 (which is responsible for the constitutive cellular uptake of iron), and therefore, we cannot exclude the potential of the HFE-like peptides to interfere with transferrin binding to TFR1. However, holo-transferrin binds TFR1 much more strongly than native HFE, providing an affinity gap that can be explored to selectively target the HFE/TFR1 interaction. In the future, it would be valuable to investigate the effect of these two families of peptides in both normal and pathological animal models. These add several layers of complexity and peptide degradation pathways and, therefore, may require more stable and potent analogues, but would provide insight into iron homeostasis in a whole-organism context.

## 4. Materials and Methods

### 4.1. General Methods

Reverse phase high performance liquid chromatography (RP-HPLC) was performed using a Shimadzu Prominence equipped with a Grace^TM^ Vydac^®^ C_18_, 250 × 21.2 mm preparative column or with a Grace^TM^ Vydac^®^ C_18_, 250 × 10 mm, semi-preparative column. Purifications involved a mobile phase of 0.05% (*v*/*v*) TFA (trifluoroacetic acid) in Milli-Q^®^ water (buffer A) mixed with 0.05% (*v*/*v*) TFA and 90% (*v*/*v*) ACN (acetonitrile, CH_3_CN) in Milli-Q^®^ water (buffer B) over a 0–80% buffer B gradient. Analytical RP-HPLC was performed using a Shimadzu Prominence equipped with a Grace^TM^ Vydac^®^ 218TP^TM^ C_18_, 150 × 2.1 mm, 5 µm, column.

Mass spectroscopy (MS) was performed using an ABSciex API 2000^TM^ coupled to an Agilent 1260 Infinity, and analytical liquid chromatography mass spectroscopy (LC-MS) was performed using the same system equipped with a Kinetex^®^ 2.6 µm C_18_ 100 Å, 50 × 2.1 mm LC column. Separations involved a mobile phase of 0.1% (*v*/*v*) FA (formic acid) in Milli-Q^®^ water (solvent A) mixed with 0.1% (*v*/*v*) FA and 90% (*v*/*v*) CH_3_CN in Milli-Q^®^ water (solvent B). The separation method comprised a 5–95% solvent B gradient (0–6 min), 95% solvent B phase (6–8 min), 95–5% solvent B gradient (8–9 min), and an equilibration phase at 5% solvent B (9–12 min).

All ^1^H NMR spectra were recorded at 298 K on a Bruker Avance 600 MHz spectrometer equipped with a cryoprobe and processed with Topspin (Bruker). Peptides were dissolved in 500 μL of H_2_O/D_2_O (450 μL/50 μL, pH 3.5) at a concentration of 1 mg/mL. For structural analysis, the homonuclear ^1^H-^1^H 2D TOCSY and NOESY experiments were recorded with mixing times of 200 ms and 80 ms, respectively, and referenced to the residual water signal at 4.768 ppm. Spectra were assigned using sequential assignment strategies [25], within CCPNMR Analysis [22]. For the identification of secondary structure, secondary Ha shifts were determined using random coil chemical shifts [21].

### 4.2. Peptide Synthesis

All peptides were synthesised on a CS Bio Co. CS336X peptide synthesiser using Rink amide MBHA resin, Fmoc amino acids, and the in situ 2-(1*H*-benzotriazol-1-yl)-1,1,3,3-tetramethyluronium hexafluorophosphate (HBTU) coupling protocol for Fmoc chemistry [26]. Double couplings were carried out for all β-branched amino acids. In brief, Fmoc amino acids (4 equiv.), HBTU (4 equiv.), and diisopropylethylamine (DIPEA) (4 equiv.) were dissolved in DMF (up to 0.2 M with respect to the amino acid), added to resin, and reacted for at least 40 min. Fmoc deprotection was carried out using 20% (*v*/*v*) piperidine in DMF (2 × 5 mL, 1 + 8 min).

For lactam-linked peptides, the incorporation of Lys^i + 4^ as Fmoc-Lys(Mtt)-OH and Asp^i^ as Fmoc-Asp(OPip)-OH was followed by the removal of side-chain protecting groups using a solution of 2% TFA in dichloromethane (10 × 1 min) [27,28]. Lactam cyclisation was subsequently achieved with a solution of 1-[Bis(dimethylamino)methylene]-1H-1,2,3-triazolo[4,5-b]pyridinium 3-oxide hexafluorophosphate (HATU) (4 equiv.) and DIPEA (4 equiv.) in DMF (up to 0.2 M) for 6 h. The remaining amino acid residues were then coupled as above.

Solid phase peptide synthesis was followed by peptide cleavage for 2 h with a solution of TFA:H_2_O:TIPS (95:2.5:2.5), after which the filtrate was collected and evaporated, and the crude peptide precipitated with ice-cold diethyl ether. The precipitate was then filtered, dissolved in 50% (*v*/*v*) buffer A/B, and lyophilised. Peptides were purified using a preparative RP-HPLC column and a 1% gradient of buffer B in buffer A. Fractions containing the linear peptide (>75%) were collected. When purity <95%, the fractions collected were repurified in a semi-preparative RP-HPLC column with a 0.6% gradient of buffer B in buffer A Appendix A.

### 4.3. In Vitro Immunofluorescence and Proximity Ligation Assay

Hepa 1–6 cells stably expressing FLAG-tagged mouse HFE were prepared and maintained as previously described [14]. Briefly, 25,000 cells were seeded, treated with peptides (50 μM), and incubated overnight before being washed with PBSCM (phosphate-buffered saline (PBS) containing 1 mM CaCl_2_ and 1 mM MgCl_2_) twice and fixed with cold 3% paraformaldehyde fixing solution (stock solution: aqueous 9% paraformaldehyde, 10x PBS, 1 M MgCl_2_, 1 M CaCl_2_) for 15 min. After rinsing with PBSCM, PBSCM containing ammonium chloride (NH_4_Cl, 50 mM) and again with PBSCM, the cells were permeabilised with 0.1% saponin in PBSCM for 15 min and were further used for either immunofluorescence (a) or proximity ligation assay (b), as described below.

For immunofluorescence (IF) studies, the cells were subsequently incubated for 1 h at room temperature with primary antibodies: rabbit anti-TfR1, 1:100 (Zymed^®^), and mouse anti-FLAG, 1:500 (Sigma-Aldrich^®^), or mouse anti-TfR1, 1:500 (abcam^®^), diluted in fluorescence dilution buffer (FDB, 5% foetal calf serum, 5% normal donkey serum, 2% bovine serum albumin in PBSCM, pH 7.6). The cells were then rinsed with PBSCM (3 × 5 min) and incubated in the dark for 1 h with secondary antibodies: donkey anti-mouse Alexa488, 1:100 (Invitrogen^TM^) and donkey anti-rabbit Alexa594, 1:100 (Invitrogen^TM^) diluted in FDB. Finally, the cells were washed with PBSCM (3 × 5 min), and the cover slips were mounted onto slides using Prolong Gold anti-fade (Molecular Probes^®^) with DAPI (4′,6-diamidino-2-phenylindole dihydrochloride).

For the proximity ligation assay (PLA), the mouse/rabbit Duolink^®^ in situ red starter kit was used according to the manufacturer’s instructions (Sigma-Aldrich). Following permeabilisation, the cells were incubated in a blocking buffer (supplied with the kit) for 2 h at 37 °C in a humidified chamber. The cells were then incubated for 2 h at room temperature with primary antibodies: rabbit anti-TfR1, 1:100 (Zymed^®^), and mouse anti-FLAG, 1:500 (Sigma-Aldrich^®^), or mouse anti-TfR1, 1:500 (abcam^®^), diluted in the antibody diluents. Later, the cells were rinsed with buffer A (supplied with the kit) twice for 10 min and incubated with the PLA probe mixture for 1 h at 37 °C in a humidified chamber. After rinsing with buffer A (2 × 10 min), the cells were incubated with the ligation reaction mixture for 1 h at 37 °C in a humidified chamber, rinsed again with buffer A (2 × 10 min), and incubated in the dark with the amplification mixture for 2 h at 37 °C in a humidified chamber. Finally, the cells were washed with buffer B (supplied with the kit) for 5 min at room temperature and 0.01x buffer B for 1 min at room temperature, and then mounted onto slides using the mounting medium supplied with the kit. Images were acquired using either a Zeiss 780 NLO, a Leica TCS SP5 or a spinning disk confocal microscope (Diskovery, Andor Technology, Belfast, UK) on a Nikon Ti-E body with a Andor iXon888 EMCCD camera. Total number of protein–protein interactions per cell was quantified using the “Find Maxima” function of FIJI, and statistical significance was determined using a one-way ANOVA with Tukey’s multiple comparison test.

## Figures and Tables

**Figure 1 molecules-27-06581-f001:**
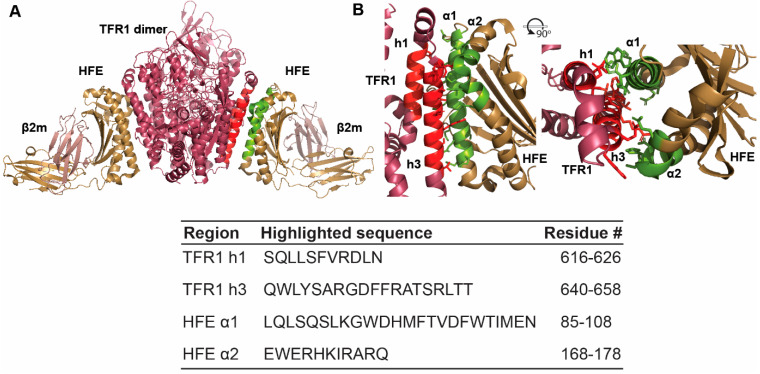
Crystal Structure of the HFE/TFR1 complex (PDB 1DE4). TFR1 is coloured in raspberry pink and HFE is coloured in sand yellow. (**A**) The full HFE/TFR1 complex and (**B**) the HFE/TFR1 interface. The complex interface involves the TFR1 three-helix bundle and HFE α-helical segments 1 and 2. The h1 and h3 correspond to the TFR1 helical segments 1 and 3, and α1 and α2 correspond to HFE α1 and α2 helices, respectively. The corresponding sequences and residue numbers for these regions are listed in the table. The interacting helical regions are coloured in red for TFR1 and in green for HFE.

**Figure 2 molecules-27-06581-f002:**
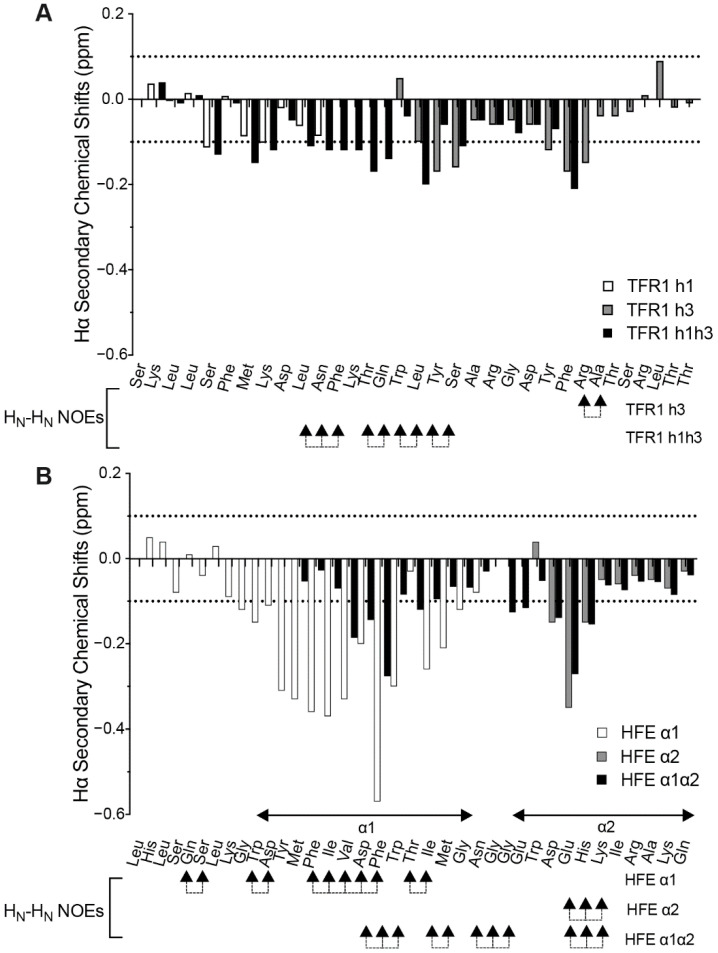
Comparison of the Hα secondary chemical shifts for both series of target peptides. All ^1^H NMR spectra were recorded at 298 K on a Bruker Avance 600 MHz spectrometer. Spectra were assigned using CCPNMR Analysis [22]. (**A**) TFR1-like series of target peptides: TFR1 h1, TFR1 h3, and TFR1 h1h3. (**B**) HFE-like series of peptides: HFE α1, HFE α2, and HFE α1α2.

**Figure 3 molecules-27-06581-f003:**
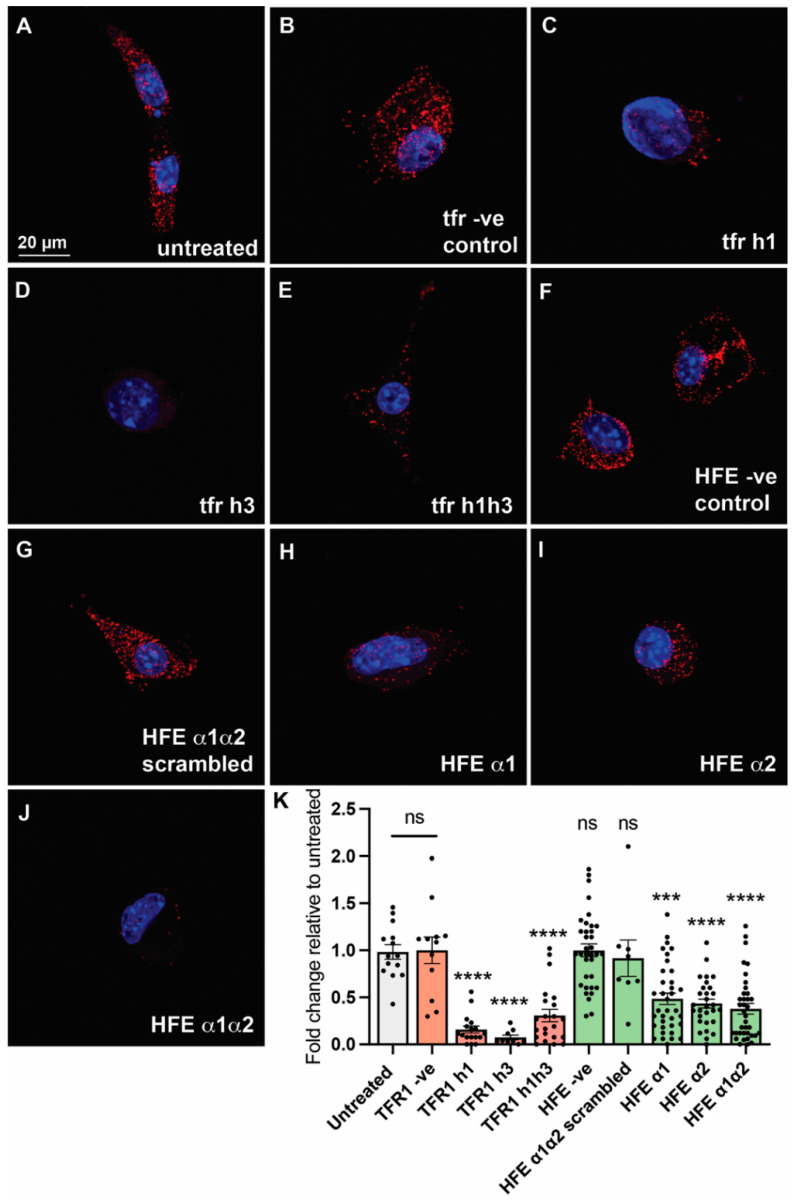
Imaging and quantification of the proximity ligation assay of HFE- and TFR1-like peptides in mouse liver Hepa 1–6 cells. The sharply bright red spots correspond to TFR1 in complex with HFE. (**A**) Untreated cells; (**B**) TFR1 negative control peptide designed from the outside of the binding interface of HFE and TFR1; (**C**–**E**) TFR1-like peptides corresponding to TFR1 h1, TFR1 h3 and TFR1 h1h3, respectively; (**F**) HFE negative control peptide designed from the outside of the binding interface of HFE and TFR1; (**G**) negative control peptide corresponding to a scrambled sequence of HFE α1α2; (**H**–**J**) HFE-like peptides corresponding to HFE α1, HFE α2, and HFE α1α2, respectively; and (**K**) bar scatter plot showing the number of red spots per cell quantified using IMARIS^®^ 8.2.0 and normalised. All peptides (other than negative control and scrambled peptides) successfully reduced the number of HFE-TFR1 complexes found per cell with respect to untreated cells (n ≥ 9. One-way ANOVA with Tukey’s test, **** *p* < 0.0001, *** *p* < 0.0002).

**Figure 4 molecules-27-06581-f004:**
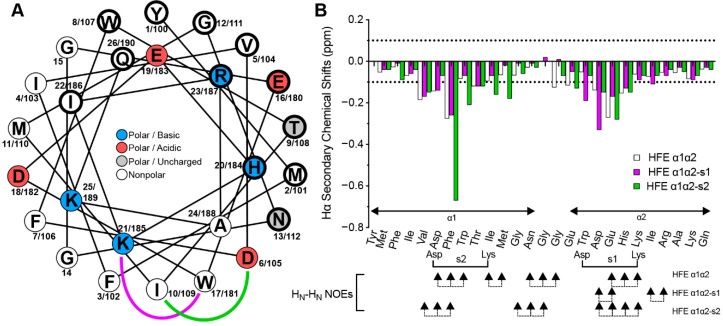
Design and structural analysis of HFE α1α2 stapled analogues. (**A**) Helical depiction of HFE α1α2 and stapled variants. Sidechain staple linkages are shown in purple (HFE α1α2-s1, stapled between Lys^21^, and an Asp introduced at position 17) and green (HFE α1α2-s2, stapled between Asp^6^ and a Lys introduced at position 10). Residues on the interaction face with Tfr are shown by bold circles. (**B**) Comparison of the Hα secondary chemical shifts for HFE α1α2 and stapled analogues 1 and 2.

**Figure 5 molecules-27-06581-f005:**
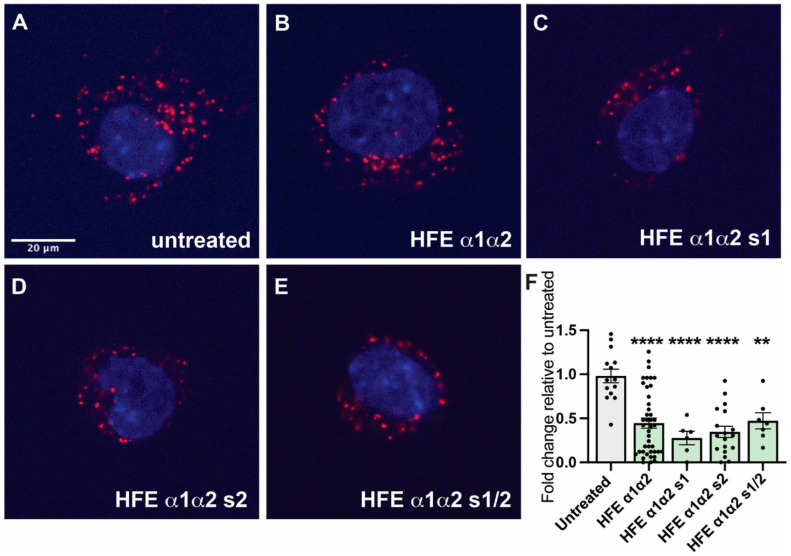
Imaging and quantification of the proximity ligation assay of the HFE α1α2 stapled peptides. Mouse liver Hepa 1–6 cells were seeded and treated with peptides (50 μM final concentration) overnight, after which they were washed, fixed, and stained according to the Duolink^®^ kit instructions. Images were acquired using either a Zeiss 780 NLO, a Leica TCS SP5 or a spinning disk confocal microscope (Diskovery, Andor technology, UK) on a Nikon Ti-E body with a Andor iXon888 EMCCD camera. Total number of protein–protein interactions per cell was quantified using the “Find Maxima” function of FIJI. (**A**) Untreated cells; (**B**) HFE α1α2; (**C**–**E**) HFE α1α2-s1, HFE α1α2-s2 and HFE α1α2-s1/s2, respectively. The sharply bright red spots correspond to TFR1/HFE complexes. (**F**) Bar scatter plot showing quantification of the proximity ligation assay. All stapled derivatives reduced the number of HFE–TFR1 complexes found per cell with respect to untreated cells but were not significantly different to the unstapled HFE α1α2 peptide. (n ≥ 6. One-way ANOVA with Tukey’s test, **** *p* < 0.0001, ** *p* < 0.002).

**Table 1 molecules-27-06581-t001:** Library of TFR1- and HFE-like peptides (based on mouse, *Mus musculus*, protein sequence). Amino acid differences from human, *Homo sapiens*, sequences are shown underlined. Sequences were written using the single-letter amino acid code. # = Number.

Peptide	Sequence	# Residues	Residue # in Protein Sequence and Comments
TFR1 h1	SKLLSFMKDLN	11	619–629
TFR1 h3	QWLYSARGDYFRATSRLTT	19	643–661
TFR1 h1h3	SKLLSFMKDLNFKTQWLYSARGDYF	25	619–629 F K^632^ T 643–653
TFR1 negative control	LKLAQVFSDMISKD	14	430–443
HFE α1	LHLSQSLKGWDYMFIVDFWTIMGN	24	89–112
HFE α2	EWDEHKIRAKQ	11	180–190
HFE α1α2	YMFIVDFWTIMGNGGEWDEHKIRAKQ	26	100–112 GG 180–190
HFE α1α2-s1	YMFIVDFWTIMGNGGEDDEHKIRAKQ	26	Lactam staple linking D^17^ to K^21^
HFE α1α2-s2	YMFIVDFWTKMGNGGEWDEHKIRAKQ	26	Lactam staple linking D^6^ to K^10^
HFE α1α2-s1/2	YMFIVDFWTKMGNGGEDDEHKIRAKQ	26	Lactam staple linking D^6^ to K^10^ and D^17^ to K^21^
HFE α1α2 scrambled	WDNRKAYMIGIETQDKFHIVMEWGFG	26	Random scrambling of the sequence of HFE α1α2
HFE negative control	EDNSTSGFWRYGYDG	15	140–153

## Data Availability

Not applicable.

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
