# Peer review of "Engineering Peptide Inhibitors of the HFE–Transferrin Receptor 1 Complex"

_molecules, 2022, doi:10.3390/molecules27196581_

Round 1

Reviewer 1 Report

This manuscript by Monteiro et al. describes the use of a proximity ligation assay to monitor disruption of the homeostatic iron regulator (HFE)-transferrin receptor 1 (TFR1) complex in mouse liver cells, using synthetic peptides corresponding to portions of the HFE or TFR1 chains. The peptides are characterized by proton NMR spectroscopy, and the effects of peptide ‘stapling’ are evaluated using PLA and NMR. The authors conclude that stapling stabilized helical structure of the inhibitors, but did not improve their potency.

The manuscript reads nicely and the key data are presented clearly in high-quality figures. Although outside my area of expertise, I expect the results would be of interest to those focused on the molecular basis of iron homeostasis, and potentially to the range of practitioners interested in using peptides/peptidomimetics to disrupt protein-protein interactions. That said, I do have some reservations about publication in current form, which should be addressable in a revision:

1) PLA provides a single measure of HFE-TFR1 complex disruption. This may be OK at an early stage, but conclusions would be strengthened by the use of an orthogonal assay (for example, a biophysical assay using purified protein domains), and this should perhaps be acknowledged. I have no direct experience with PLA, but would also ask the following:

-Is there an appropriate positive control? I cannot find one as presented, but potentially purified transferrin would be of interest.

-Is disruption of the HFE-TFR1 complex titratable, with respect to the inhibitors? As presented just a single peptide concentration is used throughout.    

2) The helix-stabilizing effect of ‘stapling’ may be overstated. For example, in Figure 4b, neither of the stapled variants appear significantly improved with respect to either: a) number of HN-HN NOEs observed; or b) number of Halpha secondary shifts < -0.1 ppm. By the later metric, the most helical peptide studied is one of the non-stapled (HFE alpha1, with 14 Halpha secondary shifts < -0.1 ppm).

3) In my opinion, the manuscript would be strengthened by placing the results in broader context, with respect to both: a) the effects of stapling on inhibitor structure and potency more generally; and b) the use of a helical peptide to disrupt a ‘discontinuous’ protein-protein interface. With respect to stapling – how do the results here compare to others’ findings? With respect to helix-based inhibitors – classic examples disrupt helix/hydrophobic groove-type interactions, where the important residues from one partner form a ‘continuous’ binding epitope (close in sequence). Is there structural precedent for the HFE/TFR1 peptide and TFR1/HFE peptide complexes implicated here?    

Other comments:

1) In Fig 1, panels A and B show the HFE-TFR1 complex from the same perspective; consider using other perspectives (e.g. top-down; possibly in additional panels) to give a more complete description of the binding interface.

2) Is there a Supporting Information document?  I don’t seem to have access to this but please ensure LC-MS analyses of synthetic peptides are included. In addition, some important experimental details could also be added to Materials and Methods section in the main text (e.g. peptide concentration used in PLA; sample conditions for NMR spectroscopy such as concentration and pH).

Reviewer 2 Report

In the current manuscript entitled “Engineering peptide inhibitors of the HFE-Transferrin Receptor 1 Complex” Clark and co-workers reported the rational design of short to medium-sized peptides as inhibitors for the interactions between HFE-TFR1 proteins. Interestingly, these rationally designed peptides were chosen from the native sequences of either protein (HEF, TFR1) sequences. The authors also synthesized mono and di-stapled peptides to increase the helicity and stability. Further, the helicities of these peptides were studied by Hα secondary chemical shifts on NMR. Later, the binding efficiencies of these peptides were studied against HEF-TFR1 interactions invitro using proximity ligation assay. Overall, the authors performed the design, synthesis, and biological evaluation of the peptides against HEF-TFR1 protein-protein interactions.

In my opinion, this manuscript should be accepted after addressing the following comments:

1.    It would be helpful to the readers if the authors mention the residual numbering for the h1 and h3 domains of TFR1 and α1 and α2 domains of HFE proteins.

2.    Why does the larger peptide (HFEα1α2) get less negative Hα secondary chemical shifts than HFEα1, which is opposite to the case for TFR1 h1h3? Do the authors have any comments?

3.    In section 2.3, the authors mentioned that TFR1 h3, TFR1 h1h3, and HFE α1α2 were found to be active peptides. But, the image (E, Fig 3) and the scattered plot (K, Fig 3) indicate that TFRh1h3 is less effective than that TFR1h1. Please address this.

4.    The citation is missing for Mtt and OPip group removal conditions, please include it.

5.    Why didn’t the authors consider the stapling of the TFR1 h3 peptide?

6.    What peptide concentrations were used? Can authors also measure the KD values on SPR or BLI?

7.    Can authors perform other stapling strategies and check the inhibition efficiency?

8.    HPLC-MS data is missing for the synthetic peptides. Can authors include them as well?

Author Response

Please see the attached pdf for our reply to reviewer 2
